# Endoscopic Ultrasonography-Guided Fine Needle Aspiration for Extrahepatic Cholangiocarcinoma: A Safe Tissue Sampling Modality

**DOI:** 10.3390/jcm8040417

**Published:** 2019-03-27

**Authors:** Takumi Onoyama, Kazuya Matsumoto, Yohei Takeda, Soichiro Kawata, Hiroki Kurumi, Hiroki Koda, Taro Yamashita, Tomoaki Takata, Hajime Isomoto

**Affiliations:** Division of Medicine and Clinical Science, Faculty of Medicine, Tottori University, Yonago, Tottori 683-8504, Japan; golf4to@yahoo.co.jp (T.O.); yhytkd7@outlook.jp (Y.T.); kawataso0527@yahoo.co.jp (S.K.); kurumi_1022_1107@yahoo.co.jp (H.K.); hkoda@aichi-cc.jp (H.K.); yamat11@gmail.com (T.Y.); t-takata@med.tottori-u.ac.jp (T.T.); isomoto@med.tottori-u.ac.jp (H.I.)

**Keywords:** EUS-FNA, ERCP, cholangiocarcinoma, accuracy, adverse event

## Abstract

Few studies have compared the diagnostic utility of endoscopic ultrasonography-guided fine needle aspiration (EUS-FNA) and endoscopic retrograde cholangiopancreatography (ERCP) tissue sampling for extrahepatic cholangiocarcinoma (ECC). We evaluated the efficacy and safety of EUS-FNA for diagnosing ECC compared with ERCP tissue sampling. Patients who underwent EUS-FNA or ERCP tissue sampling to differentiate ECC from benign biliary disease were enrolled retrospectively between October 2011 and March 2017. We evaluated diagnostic performances of EUS-FNA and ERCP tissue sampling based on pathological evaluation. We compared adverse events in EUS-FNA and ERCP tissue sampling. We enrolled 73 patients with biliary disease and performed EUS-FNA and ERCP in 19 and 54 patients, respectively. Sensitivity, specificity, and accuracy of ERCP tissue sampling were 76.0%, 100%, and 88.9%, respectively, and for EUS-FNA these were 81.8%, 87.5%, and 84.2%, respectively. Statistical values of ERCP tissue sampling and EUS-FNA were not significantly different. The adverse event frequency of EUS-FNA was significantly lower than that of ERCP tissue sampling (0% vs. 25.9%, *p* = 0.033). The diagnostic ability of EUS-FNA for ECC is similar to that of ERCP tissue sampling. EUS-FNA is a safer tissue sampling modality than ERCP for evaluating biliary disease.

## 1. Introduction

Extrahepatic cholangiocarcinoma (ECC) is a poor prognosis disease, with a five-year survival rate of 20.5% (median survival time, 11.3 months), because it is often diagnosed at an advanced stage and is often unresectable [1]. The prognosis of ECC might improve with early diagnosis of ECC [2]. It is often difficult, however, to differentiate between ECC and benign biliary strictures, such as in primary sclerosing cholangitis, IgG4-associated sclerosing cholangitis, and Mirizzi syndrome. It is important to distinguish ECC from benign biliary disease because the treatment strategies and prognoses differ. Endoscopic retrograde cholangiopancreatography (ERCP) is a common method for tissue sampling in patients with biliary strictures, using bile aspiration cytology, biliary brush cytology, and forceps biopsy. The specificity of pathological examination of tissues obtained by ERCP for biliary strictures is nearly 100%. Obtaining histological or cytological evidence is very important to determine the therapeutic strategies in these patients. The sensitivities of bile aspiration cytology, brush cytology, and forceps biopsy for biliary strictures are, however, unsatisfactory in the range of 6%–72% [3,4]. Furthermore, even though adverse events in ERCP tissue sampling are uncommon, the incidence rates of post-ERCP adverse events, which have been reported in the range of 4.0%–6.9% and include pancreatitis (2.6%–3.5%), bleeding (0.3%–1.3%), and perforation (0.1%–0.6%), cannot be ignored [5,6,7].

Recently, tissue acquisition for biliary stricture using endoscopic ultrasonography-guided fine needle aspiration (EUS-FNA) has become more common. Sadeghi et al. reported that the sensitivity and specificity of EUS-FNA for the diagnosis of malignant biliary stricture were 80% and 97%, respectively, and adverse events were rare (bleeding, 1.0% and biliary peritonitis, 0.3%) [8]. Furthermore, recent studies have reported that EUS-FNA is superior to ERCP tissue sampling in evaluating suspected malignant biliary strictures [9,10], however, the reported sensitivity of EUS-FNA for ECC (25%–100%) remains uncertain [9,10,11,12]. Moreover, there have been no studies comparing the diagnostic utility of EUS-FNA and that of ERCP tissue sampling for ECC. In addition, two issues that cannot be ignored are that EUS-FNA for biliary lesion may cause bile peritonitis and needle tract seeding [13]. In theory, the risk of bile leakage with EUS-FNA for distal biliary lesion may be lower than that for perihilar biliary lesion because the puncture route for distal biliary lesion is nearly non-transperitoneal. Considering that adverse events following EUS-FNA are rare, EUS-FNA for distal biliary lesion might be a superior diagnostic method to ERCP. In the present study, we examined the diagnostic performance and associated adverse events of EUS-FNA and ERCP tissue sampling for ECC. We also examined the utility of EUS-FNA for distal cholangiocarcinoma.

## 2. Materials and Methods

### 2.1. Study Population

In the study, 73 patients with biliary disease were enrolled at our hospital retrospectively between October 2011 and March 2017. Inclusion criteria were as follows: (1) Patients who underwent EUS-FNA and/or ERCP tissue sampling to differentiate cholangiocarcinoma from benign biliary disease; and (2) Patients aged 20 years or older when endoscopic procedures were performed. Exclusion criteria were as follows: (1) Patients who had not obtained consent; and (2) Patients who had received chemotherapy for malignant tumors within one month prior to the acquisition of pathological specimens.

Participants included 49 men and 24 women aged 42–88 years (median age, 72 years). Thirty-seven patients had ECC and 36 had benign lesion (Table 1). We measured tumor size in endoscopic ultrasonography (EUS) or computed tomography (CT). Length of stricture was measured in ERCP or CT. The EUS-FNA group was defined as the patients on whom EUS-FNA was performed as the first modality to tissue sampling for biliary lesion. The ERCP group was defined in the same way. We evaluated the diagnostic ability of EUS-FNA and that of ERCP tissue sampling for ECC on the basis of the pathological evaluation. Furthermore, we compared adverse events in EUS-FNA with those in ERCP. This study was performed according to the guidelines described in the Declaration of Helsinki, which has been developed by the World Medical Association, for Ethical Principles for Medical Research Involving Human Subjects. The study was approved by the institutional review board of Tottori University (approval number: 17A095). Informed consent was obtained from all participants using an opt-out approach in this retrospective study.

### 2.2. Endoscopic Procedure

We performed ERCP, transpapillary bile aspiration cytology and biliary forceps biopsy using a side-viewing duodenoscope (JF260V; Olympus Optical Co., Ltd., Tokyo, Japan). Bile was collected by aspiration through a biliary catheter from the bile duct during ERCP using a cannula (M00535700; Boston Scientific Corporation, Natick, MA, USA). A 0.035-inch hydrophilic guidewire (M00556051; Boston Scientific Corporation, Natick, MA, USA) and/or a 0.025-inch hydrophilic guidewire (G-260-2545A; Olympus Optical Co., Ltd., Tokyo, Japan) were also used during ERCP. The cannula was advanced into the bile duct over the guidewire. The guidewire was then withdrawn and bile was collected using a syringe with the tip of the cannula in the bile duct.

Biliary forceps biopsy was performed under X-ray fluoroscopy. The forceps were inserted into the biliary tract, opened at the perihilar side of the stricture and then drawn into the stricture. The forceps were pressed against the stricture if they became stuck and closed to carry out a biopsy of the tissue. FB-45Q-1 (Olympus Optical Co., Ltd., Tokyo, Japan) biopsy forceps, with a 2.6 mm diameter cup, were used to carry out a wire-guided biopsy. If necessary, endoscopic sphincterotomy was carried out in difficult cases to insert the forceps into the bile duct using a sphincterotome (KD-V411M-0725; Olympus Optical Co., Ltd., Tokyo, Japan).

We performed EUS-FNA using a convex linear-array echoendoscope (GF-UCT240, GF-UCT260; Olympus Optical Co., Ltd., Tokyo, Japan) connected to an ultrasound device (EU-ME2 Premier Plus; Olympus Optical Co., Ltd. and SSD-5500; Aloka, Tokyo, Japan). Puncture was performed via the duodenal wall with a 19-, 22- or 25-gauge needle (NA-U200H-8022; Olympus Optical Co., Ltd. and M00550011, M00555511; Boston Scientific Corporation, Natick, MA, USA), guided using real-time EUS imaging (Figure 1). If necessary, endoscopic biliary drainage was performed in patients with biliary stricture after EUS-FNA.

### 2.3. Diagnostic Criteria

The diagnosis of cholangiocarcinoma was based on pathological diagnosis of bile aspiration cytology, transpapillary forceps biopsy, EUS-FNA or surgical specimen. Cytodiagnosis of the specimens was performed using Papanicolaou’s method. Biopsy specimens were stained with hematoxylin and eosin and, if necessary, immunostaining, including Ki-67, p53, cytokelatin7, cytokelatin20, carcinoembryonic antigen (CEA), carbohydrate antigen 19-9 (CA19-9) and hepatocyte paraffin1, was performed. In histological findings, malignancy or suspected malignancy were considered positive (Figure 2). Patients with benign disease had a final diagnosis based on clinical and radiological follow-up data after 12 months or more.

### 2.4. Statistical Analysis

Statistical analysis was performed using StatFlex ver. 6.0 for Windows (Artech Co, Ltd., Osaka, Japan). Categorical variables were compared using the chi-square test. Continuous variables were compared by using the Mann-Whitney *U*-test. All values are expressed as means ± standard deviation or means with interquartile ranges. *p* < 0.05 was considered significant.

## 3. Results

### 3.1. Patients’ Characteristics and Baseline Evaluation

The characteristics of patients with biliary disease are shown in Table 1 and Table 2. The malignant group included 16 patients with perihilar cholangiocarcinoma and 21 with distal cholangiocarcinoma. Macroscopic types of ECC were five papillary type, 29 nodular type, and three flat type. The benign group included 25 patients with benign biliary strictures, four with IgG4-associated sclerosing cholangitis, four with pancreaticobiliary maljunction, two with intraductal papillary neoplasm of the bile duct, and one with bile duct adenoma. The final clinical diagnosis was derived from surgical pathology in 21 patients (Figure 3). We performed EUS-FNA for 19 patients and ERCP for 54 patients as the first modality to tissue sampling. In the ERCP group, all patients underwent transpapillary bile aspiration cytology and 32 patients also underwent biliary forceps biopsy. The mean number of biopsies was 3.4 (range, 1–8). In the EUS-FNA group, the mean number of needle passes was 2.1 (range, 1–3). We mainly used a 22- or 25-gauge needle (one 19-gauge, ten 22-gauge, and eight 25-gauge) in this study.

There were considerably fewer patients with perihilar biliary stricture in the EUS-FNA group than in the ERCP group. There was no significant difference in age, sex, length of stricture, size of tumor and presence of cholangitis between the EUS-FNA and ERCP groups. There were no significant differences in the median level of serum total bilirubin (T-Bil), carcinoembryonic antigen (CEA), and carbohydrate antigen 19-9 (CA19-9) between the EUS-FNA group and ERCP group.

### 3.2. Diagnostic Peformance of EUS-FNA and ERCP Tissue Sampling for Extrahepatic Cholangiocarcinoma

Table 3 summarizes the diagnostic performance of bile aspiration cytology, transpapillary biliary forceps biopsy and EUS-FNA to differentiate ECC from benign biliary disease. The values for sensitivity, specificity, positive predictive value (PPV), negative predictive value (NPV) and accuracy of ERCP tissue sampling were 76.0%, 100%, 100%, 82.9%, and 88.9%, respectively, and for EUS-FNA, the values were 81.8%, 87.5%, 90.0%, 77.8%, and 84.2%, respectively. There were no significant differences in the sensitivity, specificity, PPV, NPV and accuracy of ERCP tissue sampling and EUS-FNA for all study patients. For distal biliary lesion, the values for sensitivity, specificity, PPV, NPV, and accuracy of ERCP-based technique were 61.5%, 100%, 100%, 77.3%, and 83.3%, respectively, and for EUS-FNA, the values were 87.5%, 87.5%, 87.5%, 87.5%, and 87.5%, respectively. There were also no significant differences in the sensitivity, specificity, PPV, NPV, and accuracy of ERCP tissue sampling and EUS-FNA for patients with distal biliary lesion.

### 3.3. Adverse Events

Adverse events with ERCP and EUS-FNA are shown in Table 4. In this study, adverse events following ERCP occurred in 14 patients (25.9%). Acute pancreatitis occurred at a rate of 14.8% (8/54), with one case of severe pancreatitis. Infection occurred at a rate of 11.1% (6/54, cholangitis). All cases were resolved with conservative treatment. In contrast, there were no adverse events in the EUS-FNA group. Thus, adverse events following EUS-FNA for the evaluation of biliary disease were significantly less than those following ERCP (0% vs. 25.9%; *p* = 0.033). In addition, tumor seeding was not seen in any patients during follow-up. Serious adverse events such as perforation or hemorrhage were not observed. There was no procedure-related mortality.

## 4. Discussion

ERCP is a common method of tissue sampling in patients with biliary strictures, using bile aspiration cytology, biliary brush cytology and forceps biopsy. A recent study showed that the sensitivity and accuracy of bile aspiration cytology for malignant biliary strictures were 41.6% and 67.7%, respectively [3]. Navaneenthan et al. reported that the sensitivities of transpapillary brush cytology and forceps biopsy in diagnosing malignant biliary strictures were 45.0% and 48.1%, respectively. A combination of both modalities only modestly increased the sensitivity to 59.4%. Both techniques are almost 100% specific [4]. Although ERCP tissue sampling plays an important role in the diagnosis of biliary stricture, the sensitivity is inadequate. To improve the sensitivity for ECC, multiple sampling strategies, such as immunohistochemistry testing, mutational analysis, digital image analysis and fluorescence in situ hybridization, have been used. The specificity for diagnosis of ECC is, however, insufficient in these methods [14,15]. Furthermore, the most important problem of ERCP is the high adverse event rate, especially post-ERCP pancreatitis. Previous studies have reported incidence rates of post-ERCP adverse events of 4%.0–6.9%, including pancreatitis (2.6%–3.5%), bleeding (0.3%–1.3%), and perforation (0.1%–0.6%) [5,6].

The high sensitivity (80%) and low adverse event rate (bleeding, 1.0%; and biliary peritonitis, 0.3%) of EUS-FNA for the diagnosis of malignant biliary strictures have recently been reported [8,9]. A summary of the reported diagnostic ability of EUS-FNA for biliary lesions is shown in Table 5. Ohshima et al. reported that EUS-FNA is a sensitive (sensitivity, 100%) and safe (no adverse events) diagnostic modality for patients with suspected malignant biliary strictures after negative endoscopic transpapillary brush cytology and forceps biopsy [10]. In recent studies, EUS- FNA for biliary disease has been reported as a first-line modality for obtaining histological evidence [12,16]. Weilert at al. reported that EUS-FNA is superior to ERCP tissue sampling in evaluating suspected malignant biliary obstruction (sensitivity, 94% vs. 50%), especially, for pancreatic ductal adenocarcinoma (sensitivity, 100% vs. 38%). On the other hand, the sensitivity of EUS-FNA for ECC is similar to that of ERCP tissue sampling (79% sensitivity for both) [9]. That study is the only prospective study to compare EUS-FNA and ERCP tissue sampling for ECC. All EUS and ERCP procedures were, however, performed during a single session in that study, such that the adverse events of each procedure could not be compared. In the current study, the sensitivity of EUS-FNA for ECC was also similar to that of ERCP tissue sampling. Moreover, the safety of EUS-FNA for biliary disease was superior to ERCP. Recent studies have also reported that the adverse event frequency of EUS-FNA for ECC is nearly zero [9,10,11,12,16,17,18,19,20,21,22,23,24,25].

Small lesions, especially those present with biliary wall thickening, are more difficult to sample using EUS-FNA. Indeed, we did not perform EUS-FNA for biliary stricture without mass formation in this study. ERCP tissue sampling might be superior to EUS-FNA for ECC without mass formation.

In addition, the possibility of needle track seeding in resectable cases remains unresolved, although EI Chafic et al. reported that preoperative EUS-FNA in patients with perihilar cholangiocarcinoma did not appear to adversely affect overall or progression-free survival [13,26]. In theory, the puncture route for distal biliary lesion (transduodenal) is almost non-transperitoneal, such that there may be less bile leakage with EUS-FNA for distal biliary lesion than for perihilar biliary lesion. Moreover, previous studies have reported that the sensitivity of EUS-FNA for distal cholangiocarcinoma was higher than that for perihilar cholangiocarcinoma [12,23]. A summary of the diagnostic ability of EUS-FNA for distal biliary lesions is shown in Table 6.

Superficial intraductal spread, in which epithelium extends continuously from the main tumor, is a feature of cholangiocarcinoma [27]. The presence of superficial intraductal spread is related to positive resection margins after surgery. Therefore, preoperative identification of the exact perihilar and distal margins of resectable cholangiocarcinoma is important. The frequency of superficial intraductal spread in cholangiocarcinoma, reported as 14.6% [28], cannot be determined using EUS-FNA. Hijioka et al. reported, however, that the diagnostic yield of mapping biopsy procedures (ERCP tissue sampling) to accurately distinguish between benign and malignant foci was 89% [29]. ERCP-related forceps biopsy for defining longitudinal extension is an indispensable modality for patients with resectable ECC.

Another advantage of EUS-FNA as a first-line modality for biliary disease is that if the results of EUS-FNA show malignancy, forceps biopsy of the tumor site can be omitted in mapping biopsy during ERCP for resectable ECC. This may lead to a reduced risk of post-ERCP adverse events. Moreover, a self-expandable metallic stent, which has longer patency, can be placed during the initial ERCP for biliary drainage [30]. Due to fewer adverse events, it might be acceptable to use EUS-FNA as a first-line technique before ERCP in the diagnosis of biliary disease, especially in unresectable cases. In resectable cases, considering the possibility of needle tract seeding with EUS-FNA, distal biliary lesion with mass formation might be a first-line diagnostic modality.

The present study has some limitations. First, this was a retrospective, single-center study with a small number of cases. Second, we could not compare the diagnostic performance of EUS-FNA and that of ERCP tissue sampling with each T category. Third, patients who underwent clinical follow-up were included in this study. A prospective randomized study including a larger number of patients is needed. In addition, to evaluate the risk of needle tract seeding in EUS-FNA, long-term follow-up is needed.

## 5. Conclusions

The diagnostic ability of EUS-FNA for ECC is similar to that of ERCP tissue sampling. EUS-FNA is a safer tissue sampling modality than ERCP for the evaluation of extrahepatic biliary disease.

## Figures and Tables

**Figure 1 jcm-08-00417-f001:**
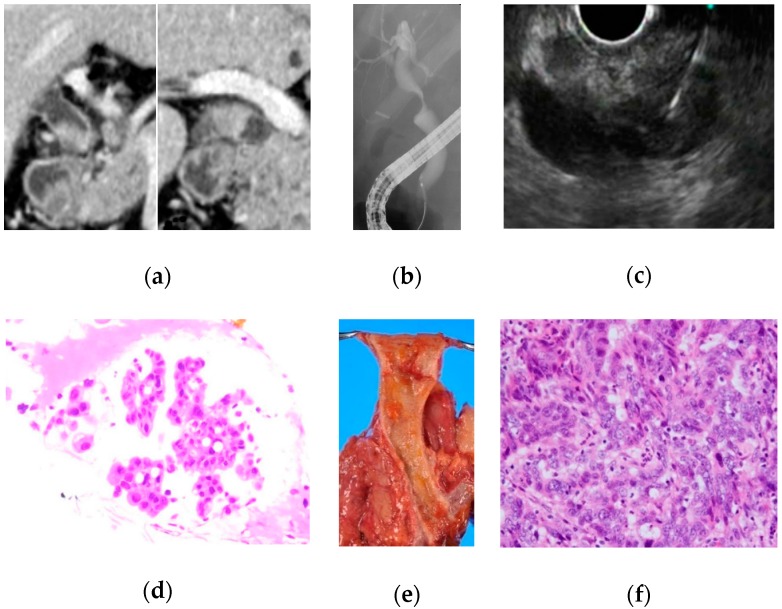
A case of bile duct stricture diagnosed as distal cholangiocarcinoma with endoscopic ultrasonography-guided fine needle aspiration (EUS-FNA). (**a**) Computed tomography (CT) scan showed an irregular nodule in the distal bile duct; (**b**) endoscopic retrograde cholangiography showed stenosis in the distal bile duct; (**c**) EUS-FNA was performed for the nodule in the distal bile duct from the duodenum; (**d**) hematoxylin and eosin staining revealed adenocarcinoma; (**e**) pancreatoduodenectomy was performed; and (**f**) histologically, this patient was diagnosed with distal cholangiocarcinoma.

**Figure 2 jcm-08-00417-f002:**
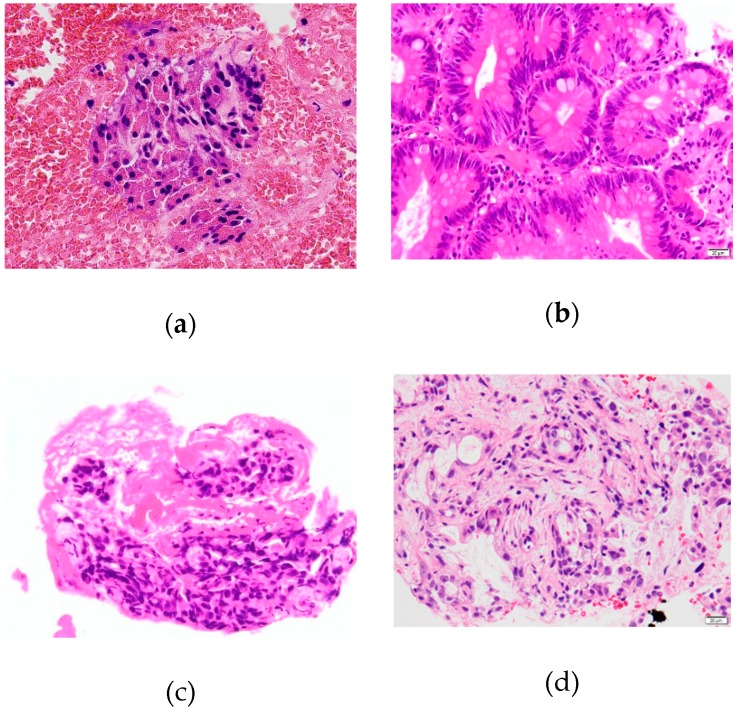
Histopathological findings of specimens obtained via EUS-FNA with normal, adenoma, suspected adenocarcinoma and adenocarcinoma. (**a**) Biliary epithelium without atypia was shown (normal); (**b**) biliary epithelial cells with mild cellular atypia were revealed (adenoma); (**c**) small amount of biliary epithelial cells with severe atypia was shown (suspected adenocarcinoma); and (**d**) biliary epithelium with severe atypia and architectural distortion was shown (adenocarcinoma).

**Figure 3 jcm-08-00417-f003:**
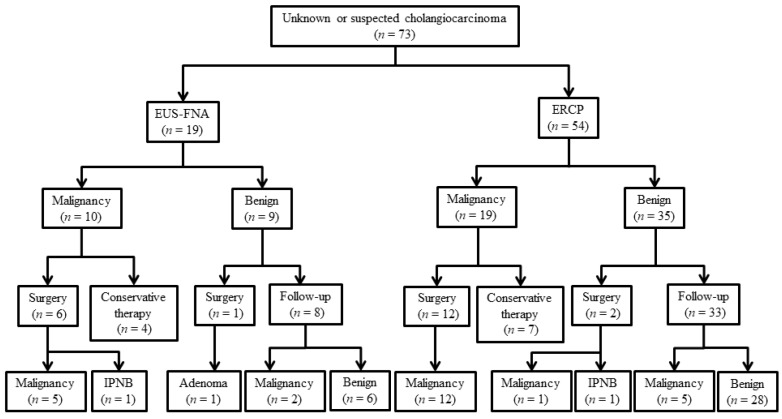
Diagnostic flowchart of patients included in the study. Abbreviation: IPBN, intraductal papillary neoplasm of the bile duct.

**Table 1 jcm-08-00417-t001:** Baseline characteristics of study patients.

	Biliary Disease (*n* = 73)
Age (range), years	72 (42–88)
Sex, male/female	49/24
Location of stricture (perihilar/distal) Malignant Benign	27/46 16/21 11/25
Length of stricture, mm	16.9 (1.2–46.0)
Cholangitis (with/without)	6/67
Total bilirubin, mg/dL	1.2 (0.3–26.2)
Tumor marker CEA, ng/mL CA19-9, U/mL	2.6 (0.8–1549.1) 37.3 (0.8–11985)
Malignant Cholangiocarcinoma	37 37
Benign Benign biliary stricture IgG4-associated sclerosing cholangitis Pancreaticobiliary maljunction Intraductal papillary neoplasm of bile duct Bile duct adenoma	36 25 4 4 2 1

Values are presented as number or median (range).

**Table 2 jcm-08-00417-t002:** Characteristics of patients with biliary disease.

	EUS-FNA (*n* = 19)	ERCP (*n* = 54)	*p* Value
Age, years	71 (59–83)	72 (42–88)	0.910 *
Sex, male/female	12/7	37/17	0.669 **
Malignancy / Benign	11/8	26/28	0.465 **
Location (perihilar/distal)	3/16	24/30	0.026 **
Length of stricture, mm	19.0 (5.4–34.4)	16.5 (1.2–46.0)	0.734 *
Size of tumor, mm	24.2 (13.5–72.0)	26.0 (0.7–80.0)	1.000 *
Cholangitis (with/without)	2/17	4/50	0.952 **
Total bilirubin, mg/dL	1.3 (0.5–25.4)	1.0 (0.3–26.2)	0.062 *
Tumor marker (serum) CEA, ng/mL CA19-9, U/mL	2.5 (0.8–8.3) 26.0 (0.8–11985.0)	2.8 (0.9–1549.1) 43.5 (0.8–4939.3)	0.777 * 0.195 *

Values are presented as number or median (range). Abbreviations: EUS-FNA, endoscopic ultrasonography-guided fine needle aspiration; and ERCP, endoscopic retrograde cholangiopancreatography. * *p* value: Mann-Whitney *U* test. ** *p* value: Chi-square test.

**Table 3 jcm-08-00417-t003:** Diagnostic ability of ERCP tissue sampling and EUS-FNA for differentiating extrahepatic cholangiocarcinoma (ECC) from benign biliary disease.

	Sensitivity, %	Specificity, %	PPV, %	NPV, %	Accuracy, %
Bile aspiration cytology	60.0 (15/25)	100 (29/29)	100 (15/15)	74.4 (29/39)	81.5 (44/54)
Forceps biopsy	75.0 (15/20)	100 (13/13)	100 (15/15)	72.2 (13/18)	84.8 (28/33)
ERCPTissue sampling	76.0 (19/25)	100 (29/29)	100 (19/19)	82.9 (29/35)	88.9 (48/54)
EUS-FNA	81.8 (9/11)	87.5 (7/8)	90 (9/10)	77.8 (7/9)	84.2 (16/19)

Abbreviations: PPV, positive predictive value and NPV, negative predictive value. *p* value: Chi-square test.

**Table 4 jcm-08-00417-t004:** Adverse events with ERCP and EUS-FNA.

Adverse Event	ERCP (*n* = 54)	EUS-FNA (*n* = 19)	*p* Value
Pancreatitis	14.8 (8/54)	0	0.177
Bleeding	0	0	NS
Infection	11.1 (6/54)	0	0.302
Perforation	0	0	NS
Cardiac	0	0	NS
Pulmonary	0	0	NS
Medication reaction	0	0	NS
Other	0	0	NS
Overall	25.9 (14/54)	0	0.033

Abbreviation: NS, not significant. *p* value: Chi-square test.

**Table 5 jcm-08-00417-t005:** Summary of the diagnostic ability of EUS-FNA for differentiating extrahepatic cholangiocarcinoma from benign biliary disease.

Author (Year)	No. Cases	Sensitivity, %	Specificity, %	Accuracy, %	Adverse Event, %
Fritscher-Ravens (2004) [17]	44	89	100	91	0
Byrne (2004) [18]	35	63	100	83	0
Rösch (2004) [11]	11 ^a^	25	100	NA	0
Lee (2004) [19]	28	47	100	NA	0
Eloubeidi (2006) [20]	25	86	100	88	0
DeWitt (2006) [21]	24	77	100	79	0
Meara (2006) [22]	43	87	100	91	0
Mohanmadnejad (2011) [23]	74	73	NA	NA	1, hemobilia
Ohshima (2011) [10]	22	100	100	100	0
Nayar (2012) [24]	32	60	100	74	0
Krishna (2012) [25]	28	67	100	79	0
Weilert (2014) [9]	15 ^a^	79	NA	80	0
Téllez-Ávila (2014) [16]	39	79	100	82	0
Onda (2016) [12]	47	84	100	87	0
Our study	19	82	88	84	0

^a^ Patients with pancreatic cancer were excluded.

**Table 6 jcm-08-00417-t006:** Summary of the diagnostic ability of EUS-FNA for differentiating distal cholangiocarcinoma from benign biliary disease.

Author (Year)	No. Cases	Sensitivity, %	Specificity, %	Accuracy, %
Byrne (2004) [18]	35	63	100	83
Lee (2004) [19]	28	47	100	NA
Mohanmadnejad (2011) [23]	51 ^a^	81	NA	NA
Ohshima (2011) [10]	13 ^a^	100	100	100
Nayar (2012) [24]	32	60	100	74
Weilert (2014) [9]	8 ^a^	71	100	75
Onda (2016) [12]	21 ^a^	89	100	90
Our study	16	88	88	88

^a^ Patients with perihilar cholangiocarcinoma and pancreatic cancer were excluded.

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
