# Peer review of "Endoscopic Ultrasonography-Guided Fine Needle Aspiration for Extrahepatic Cholangiocarcinoma: A Safe Tissue Sampling Modality"

_jcm, 2019, doi:10.3390/jcm8040417_

Reviewer 1 Report

The authors presented a very interesting paper around the usefulness of EUS-FNA vs ERCP in cholangiocarcinoma patients. The number of patients enrolled is quite large and the description of procedure used is clear. The author described also the flow-chart of diagnostic process reporting also the description of the activities performed in their surgical pathology unit. Furthermore the authors mentioned the immunostaining approach iin order to diagnose malignant or benign lesions, but they didn't mention any marker (i.e. CKs). We are yhinking that this type of analyses should be mandatory in both cytological and histological diagnoses. Indeed, we would like to suggest to the authors to enhance the part of histopathological diagnosis, reporting the score/values of markers used and (very important) some picture of IHC staining.

Author Response

Dear reviewer.

Thank you for your comments and suggestions about our manuscript.

We will send you to revised manuscript.

We provide response to the following:

#. The authors mentioned the immunostaining approach in order to diagnose malignant or benign lesions, but they didn't mention any marker (i.e. CKs).

Lines 115 - 118.

As you have pointed out, we would like to insert immunostaining markers. So, we have changed the sentence “Biopsy specimens were stained with hematoxylin and eosin and, if necessary, immunostaining, including Ki-67, p53, cytokelatin7, cytokelatin20, carcinoembryonic antigen (CEA), carbohydrate antigen 19-9 (CA19-9) and Hepatocyte paraffin1, was performed.”

#. We would like to suggest to the authors to enhance the part of histopathological diagnosis, reporting the score/values of markers used and (very important) some picture of IHC staining.

In EUS-FNA group, we performed immunostaining for only a specimen obtained from a patient with perihilar cholangiocarcinoma in order to differentiate cholangiocarcinoma from hepatocellular carcinoma. CK7, CK20, CEA, CA19-9 and HEP-PAR1 were stained for this case. In ERCP group, we performed immunostaining, including p53 and Ki-67, for specimens obtained by forceps biopsy from a few patients with cholangiocarcinoma in an auxiliary diagnosis. However, all patients were diagnosed malignancy or suspected malignancy by histological findings of specimen stained with hematoxylin and eosin. As described above, we think that the picture of IHC staining might be meaningless in the distinction between malignancy and benign in our study. Moreover, we could not present the score/values of IHC staining. Specifically, what is the score/values of makers?

I hope it will be accepted for publishing in J Clin Med.

Sincerely,

Takumi Onoyama.

Division of Medicine and Clinical Science, Department of Multidisciplinary Internal Medicine, Faculty of Medicine, Tottori University, 

Reviewer 2 Report

The authors describe a retrospective analysis comparing EUS-FNA for Extrahepatic Cholangiocarcinoma with ERCP and show that the diagnostic sensitivity is similar but EUS-FNA has less adverse events including pancreatitis. 

The study is well designed and presented, some minor edits/corrections could be made (example line 46). 

Perhaps mentioning the use of EUS-FNA to diagnose other cancers (pancreatic) could build a stronger case.

Author Response

Dear reviewer.

Thank you for your comments and suggestions about our manuscript.

We will send you to revised manuscript.

We provide response to the following:

#. Some minor edits/corrections could be made (example line 46). 

Lines 43 - 45

As you have pointed out, this sentence was wrong. We have changed the sentence “Furthermore, even though adverse events in ERCP tissue sampling are uncommon, the incidence rates of post-ERCP adverse events, which have been reported in the range of 4.0–6.9% and include pancreatitis (2.6–3.5%), bleeding (0.3–1.3%), and perforation (0.1–0.6%), cannot be ignored.”

#. Mentioning the use of EUS-FNA to diagnose other cancers (pancreatic) could build a stronger case. 

Lines 197 - 201.

As you have pointed out, we have added the sentence about the usefulness of EUS-FNA to diagnose pancreatic ductal adenocarcinoma in discussion. We have changed the sentences “Weilert at al. reported that EUS-FNA is superior to ERCP tissue sampling in evaluating suspected malignant biliary obstruction (sensitivity, 94% vs 50%), especially, for pancreatic ductal adenocarcinoma (sensitivity, 100% vs 38%). On the other hand, the sensitivity of EUS-FNA for ECC is similar to that of ERCP tissue sampling (79% sensitivity for both).”

I believe the revised manuscript has been improved satisfactory and hope it will be accepted for publishing in J Clin Med.

Sincerely,

Takumi Onoyama.

Division of Medicine and Clinical Science, Department of Multidisciplinary Internal Medicine, Faculty of Medicine, Tottori University, 

Round  2

Reviewer 1 Report

Looking at the R1 version of Manuscript, the authors reported some IHC tests done, but they would like to avoid their representation by the pictures.
Furthermore, in our reply, they reported the role of IHC markers for differential diagnosis criteria: “ In EUS-FNA group, we performed immunostaining for only a specimen obtained from a patient with perihilar cholangiocarcinoma in order to differentiate cholangiocarcinoma from hepatocellular carcinoma.”

This fact seems not to be in agreement with the title of the paper, in which the authors would like to differentiate malignant vs benign lesions. Furthermore, in order to conclude this work, at least two pictures (showing negative and positive markers value), may increase the description of differential diagnosis criteria used.

Author Response

Dear reviewer.

Thank you for your careful comments and suggestions about our manuscript.

We will send you to revised manuscript.

We provide response to the following:

#. In order to conclude this work, at least two pictures (showing negative and positive markers value), may increase the description of differential diagnosis criteria used.

Lines 119 – 120, 140, 154.

As you have pointed out, we would like to insert Figure 2. As we told you now, we have not performed IHC staining for specimens obtained via EUS-FNA in order to differentiate malignant vs benign lesions. Therefore, we have shown the histological findings with normal, adenoma, suspected adenocarcinoma, and adenocarcinoma in specimens stained with hematoxylin and eosin. Moreover, we have changed our previous Figure 2 to Figure 3.

We believe the revised manuscript has been improved satisfactory and hope it will be accepted for publishing in J Clin Med.

Sincerely,

Takumi Onoyama.

Division of Medicine and Clinical Science, Department of Multidisciplinary Internal Medicine, Faculty of Medicine, Tottori University.